# The Double-Edged Sword of Beta2-Microglobulin in Antibacterial Properties and Amyloid Fibril-Mediated Cytotoxicity

**DOI:** 10.3390/ijms22126330

**Published:** 2021-06-13

**Authors:** Shean-Jaw Chiou, Huey-Jiun Ko, Chi-Ching Hwang, Yi-Ren Hong

**Affiliations:** 1Department of Biochemistry, Faculty of Medicine, College of Medicine, Kaohsiung Medical University, Kaohsiung 807, Taiwan; o870391@yahoo.com.tw (H.-J.K.); cchwang@kmu.edu.tw (C.-C.H.); 2Department of Medical Research, Kaohsiung Medical University Hospital, Kaohsiung 807, Taiwan; 3Graduate Institute of Medicine, College of Medicine, Kaohsiung Medical University, Kaohsiung 807, Taiwan; 4Department of Biological Sciences, National Sun Yat-Sen University, Kaohsiung 804, Taiwan

**Keywords:** beta2-microglobulin, antimicrobial peptide, amyloid fibrils, cytotoxicity

## Abstract

Beta2-microglobulin (B2M) a key component of major histocompatibility complex class I molecules, which aid cytotoxic T-lymphocyte (CTL) immune response. However, the majority of studies of B2M have focused only on amyloid fibrils in pathogenesis to the neglect of its role of antimicrobial activity. Indeed, B2M also plays an important role in innate defense and does not only function as an adjuvant for CTL response. A previous study discovered that human aggregated B2M binds the surface protein structure in *Streptococci*, and a similar study revealed that sB2M-9, derived from native B2M, functions as an antibacterial chemokine that binds *Staphylococcus aureus*. An investigation of sB2M-9 exhibiting an early lymphocyte recruitment in the human respiratory epithelium with bacterial challenge may uncover previously unrecognized aspects of B2M in the body’s innate defense against *Mycobactrium tuberculosis*. B2M possesses antimicrobial activity that operates primarily under pH-dependent acidic conditions at which B2M and fragmented B2M may become a nucleus seed that triggers self-aggregation into distinct states, such as oligomers and amyloid fibrils. Modified B2M can act as an antimicrobial peptide (AMP) against a wide range of microbes. Specifically, these AMPs disrupt microbe membranes, a feature similar to that of amyloid fibril mediated cytotoxicity toward eukaryotes. This study investigated two similar but nonidentical effects of B2M: the physiological role of B2M, in which it potentially acts against microbes in innate defense and the role of B2M in amyloid fibrils, in which it disrupts the membrane of pathological cells. Moreover, we explored the pH-governing antibacterial activity of B2M and acidic pH mediated B2M amyloid fibrils underlying such cytotoxicity.

## 1. Background of Beta-2 Microglobulin

Human beta-2 microglobulin (*B2M*), a small gene composed of four exons at chromo-some 15q21.1, encodes a full-length, 119-residue protein that becomes a 99-residue mature B2M (pI = 6.02) with a secondary structure of seven beta-pleated sheets (https://www.ncbi.nlm.nih.gov/gene/567 accessed on 11 May 2021). *B2M* is homologous in sequence with the homology region of IgG and HLA, indicating a common evolutionary origin [1]. B2M forms a noncovalently-linked light chain of major histocompatibility complex (MHC) class I molecules on the surface of almost nucleated cells. B2M is characterized as a scaffolding protein that retains the native structure of MHC class I molecules in antigen present cells (APCs) to present peptides to cytotoxic CD8+ T cells [2,3,4]. Because MHC class I molecules can affect many physiological functions in mice and human beings, animal models of an engineered B2M deficiency with low MHC class I levels address major challenges in the endogenous cross-presentation of the host [5,6,7]. In addition to being a part of non-membrane-bound proteins associated with MHC class I molecules, B2M can be released in free form in serum. However, the quantity of B2M in serum is dynamically related to a wide range of diseases. Although serum B2M level is insufficiently specific for use in the diagnosis of certain diseases, it is related to a high tumor burden and the poor prognosis of various cancers [8,9]. Furthermore, B2M with a predominantly beta-pleated sheet can form molecular aggregation associated with dialysis-related amyloidosis (DRA) [10,11], a feature of the deposition of amyloid fibrils in the skeletal muscular system. In particular, the systemic accumulation of B2M has been recognized a pro-aging factor that impairs cognitive function and neurogenesis [12]. B2M shedding, either intact, fragmented or in the form of amyloid fibrils, may be disastrous for the host or their generation represents an alternative defense against diseases, which is discussed as below.

## 2. Discovery of B2M’s Antimicrobial Activity

The role of B2M in adaptive immunity has been extensively studied, but its role in innate defense has been neglected. Roch et al. discovered that five earthworm defense proteins exhibited compositional similarity with mouse B2M, reflecting an evolutionary function of B2M in innate defense among invertebrates [13]. The antibacterial activity of B2M was first discovered in human amniotic fluid a decade ago [14], indicating that potassium ions are critical for the antibacterial activity of B2M. B2M causes the dissipation of bacterial transmembrane potential without damaging the cell membrane and it exhibits a wide range of antibacterial activity against not only antibiotic-susceptible but also antibiotic-resistant bacteria, including *L. monocytogenes*, *E. coli*, and *S. aureus* [14]. Due to its antigen presentation role in APCs, B2M-knockout in animal models indicates that immune cells, except for those carrying the T-cell receptor type, can also play an important role against pathogenic infection [15]. In addition, D’Souza et al. discovered that a non-class B2M-restricted mechanism affects the early accumulation of lymphocytes and resistance to *Mycobactrium tuberculosis* (*M. tuberculosis*) in mice. They also observed that mice had a ability to fight pulmonary infection with tuberculosis when this mechanism was absent [16]. Rolph et al. also demonstrated the effect that MHC class Ia-restricted T cells only partially explained B2M-dependent resistance to *M. tuberculosis* [17]. The heterogeneous nature of CD8+ T cells therefore accounts for the cellular immune response that occurs during *M. tuberculosis* infection. Furthermore, Cogen et al. also suggested that an unknown mechanism exists in B2M-dependent bacterial clearance and survival during infection by murine Klebsiella pneumoniae bacteremia [18]. These findings jointly suggest that a B2M-dependent mechanism instead of a CD8 T-cell and iNK T-cell dependent mechanism is involved in innate defense during bacterial infection in mice. B2M amyloid was found to be the most distinctive feature of deposits in affected tissues, with the adjacent chronic inflammation indicating infiltration by monocytes or macrophages. This suggests that infiltration of monocytes or macrophages is mediated by the late occurrence of B2M amyloidosis [19]. The alternative role of B2M in monocyte filtration had not been further investigated until the discovery that B2M acts as a precursor of antibacterial chemokine in human respiratory epithelial cells [20]. A 9-kDa fragment derived from B2M, named sB2M-9, was identified in the culture medium of human respiratory epithelial cells (REC) upon IL-1β stimulation and presents in the nasal fluid from healthy individuals [20]. The mechanism underlying the shedding of sB2M-9 from REC remains unknown.

Many functional peptides derived from their precursor sequences are known to be released through a proteolytic mechanism. For example, hCAP18, a human antimicrobial peptide (AMP) precursor, harbors an active domain, namely LL-37 in its C-terminal sequence [21]. A recent study found that a carboxyl terminal fragment of beta-subunit hemoglobin exhibits antimicrobial activity in the placenta for protection against bacterial and viral infection [22]. The mechanism of sB2M-9 released from its precursor B2M may be similar to those of other AMPs under limited proteolysis in response to pathogenic stimulation. sB2M-9 is a binding factor that initially binds onto the membrane surface of aggregated *S. aureus.* sB2M-9 and other defense molecules, but not native B2M, possibly account for the formation of bacterial clumps. However, these bacterial clumps last only for a few hours. A subsequent abrupt increased in the growth of the bacteria (relative to that of a control group) indicates that sB2M-9 realizes its antibacterial activity by temporarily inhibiting the growth of bacteria and not by killing them. In addition, the bacterial clumps enhance THP-1 monocyte migration, to a level that is greater than even that of bacteria migration itself without forming clumps, indicating the role of sB2M-9 as a chemokine. Thus, sB2M-9 is suggested to exhibit early lymphocyte recruitment in the milieu of REC with bacterial infection, in addition to the classic B2M-restricted mechanism; moreover, it may be responsible for resistance to tuberculosis in the lungs. Such evidence indicates that hitherto unrecognized aspects of B2M in the innate defense against *M. tuberculosis* in the lungs.

Cole et al. characterized the antibacterial activity of a panel of cationic polypeptides in human airway fluid, and they discovered that B2M plays a putative role in the suppression of *E. coli* and *L. monocytogenes* [23]. Their analysis of the antimicrobial activity of B2M was conducted in a two-dimensional gel composed of AU-PAGE in the first dimension and SDS-PAGE in the second dimension followed by a gel-overlay assay. Their findings revealed an antibacterial signal of 7- to 10 kDa, as shown on the gel, in which another AMP of 94 amino acids (i.e., Beta-microseminoprotein) was the control [24]. This signal was later identified to be from B2M through N-terminal Edman peptide sequencing and MALDI-TOF mass spectrometry. The results indicated that antimicrobial B2M is derived from intact B2M through a cleaving mechanism. Thus, the findings from both N-terminal Edman sequencing and mass spectrometry, which provide only the partial protein sequences for database search must be verified further and interpreted with caution. 

In a recent study, Holch et al. demonstrated that B2M exerts pH-dependent antimicrobial activity against *B. subtilis*, *P. aeruginosa,* and *L. monocytogenes* in human broncho-alveolar-lavage fluid [25]. This finding indicates that the antimicrobial activity of B2M localizes to its C-terminal domain, a region responsible for the formation of fibrils. In that study, the antimicrobial activity of B2M increased as pH decreased for pH < 5.5, but decreased as pH increased for pH > 6, suggesting that B2M exhibits antimicrobial activity under acidic conditions. Therefore, we can infer that an increased dose of B2M at pH 4.5 mediates the formation of fibrils and may thus contribute to the AMP effect; infection is typically associated with a low pH environment. That study demonstrated that 1 mg/mL B2M disrupts the membrane of bacterial in *L. monocytognes*. Notably, as indicated by synthetic peptides in a bacterial assay, the AMP activity of B2M is located within a fragment of 18 amino acids at the C-terminal region (i.e., B2M_99-116). In contrast to the structure of sB2M-9, as reported by our previous study [20], this 9 kDa fragment is composed of two discontinuous chains that are linked with a disulfide bond. The short chain sequence (B2M_99-111) of sB2M-9 has a near match with the 18-amino acid fragment (B2M_99-116) at the C-terminal region, as reported by Holch et al. According to pI/Mw calculations (https://web.expasy.org/compute_pi/ accessed on 11 May 2021), the theoretical pI values for these two deduced small fragments of B2M are similar and fairly high (Figure 1). 

In addition, a panel of chemokines including CXC and CC chemokines, has been characterized as exhibiting antimicrobial activity [26]. Moreover, chemokine-derived peptides, such as CXCL8_80–99_ but not intact CXCL8, were reported to exhibit antimicrobial activity [27]. This finding implies that the proteolytic process is required to shed CXCL8_80–99_ as a true AMP. In a recent study, Wendler et al. also reported that the proteolytic cleavage of reduced human beta defensin 1 (hBD-1) releases an octapeptide with antimicrobial activity [28]. AMP sequences are typically composed of fewer than 50 amino acids with a net positive charge domain and a hydrophobic domain. This feature of AMPs is believed to indicate that the AMP precursor may be subject to a proteolysis-triggered shedding mechanism [29]. The shedding process allows AMPs to retain an excess of basic amino residues while removing acidic ones under distinctly pathological conditions or physiological pH conditions. 

We recently analyzed a panel of cationic peptides cultured in A549 medium infected with IL-1β by proteomic approach, thioredoxin was especially prominent in addition to some other immune responsible factors, such as cytokines and chemokines (Table 1). Thioredoxin, a small redox protein and a chemokine [30], has been discovered to regulate a spectrum of physiological functions, including influencing a disulfide bond-linked dimer of Tapsin:Rep57 within the MHC class I peptide-loading complex [31]. In a recent study, the substantially enhanced antimicrobial activity of hBD-1 due to a reduction of disulfide bonds also indicated the key role of thioredoxin in innate defense [28]. In summary, considering the evidence for the antimicrobial activity of B2M_99-116 fragment, this study hypothesized that the short chain peptide B2M_99-111 derived from sB2M-9 plays a role in host defense as an AMP. Thus, whether an unrecognized reduction reaction by thioredoxin sheds the short chain fragment from sB2M-9 warrants investigation. 

## 3. pH-Dependent Antimicrobial Activity of B2M

AMPs can have a pH dependent mode of action in a wide range of multicellular organisms [42]. In general, they exhibit antimicrobial activity through interacting with the component of the cell membrane to disrupt the membrane’s integrity. After inserting itself into the lipid bilayer of the membrane, AMPs self-aggregate to form pores on the membrane to kill the bacterium [43]. A different group of AMPs exhibited optimal activity against microbes at acidic pH conditions; however, some of them were activated at a basic range (Table 2). The different pH values in different tissues were therefore considered to function as a natural barrier against microbial infection. The change in the pH values was thought to be involved in the control of enzyme activities that produce functional AMPs. These AMPs carry the net positive charge by protonating the amino acid residues, such as histidine, aspartic acid, and glutamic acid. Thus, the pH value of the microenvironment plays a role in the defense against and prevention of diseases. For example, the pH value of airway surface liquid (ASL) measured in vivo is approximately 6.6 in a normal human body. Similarly, alveolar subphase fluid (AVSF) had a pH of approximately 6.9 [44]. The pH of ASL and AVSF can be altered in response to infection and inflammation. In a study of the effect of ASL pH on cystic fibrosis (CF) and the polarized normal of primary bronchial epithelial cells, ASL had a lower pH value relative to the normal control [45]. The dysregulation of ASL pH, an example of how CF lacks functional CF transmembrane conductance regulator (CFTR) and promotes ASL acidification, is speculated to suppress microbe clearance. Furthermore, a study observed that the bactericidal activity of defensin in human airway epithelial cells was lower under a hyperacidification condition (pH: 3–5) [46]. The mechanism underlying an acidic pH is thought to mediate the synthesis of AMPs and alternatively trigger the degradation of AMP through the activation of host proteases. In addition, Tipping et al. demonstrated that mild acidification is a key factor to inducing the formation of fibril-derived nonnative B2M oligomers that cause membrane disruption [47]. Hsp70, a molecular chaperone, can impair B2M fibril-mediated membrane disruption. This finding implies that pH plays a role in B2M fibril stability. Thus, it is believed that the B2M fibril-induced effect on membranes is modulated by anionic lipids and enhanced at low pH values. A mechanism involving amyloid fibrils in relation to the cytotoxic effect was also indicated by the presence of endocytosed B2M amyloid fibrils that induce necrosis and apoptosis through the disruption of endosomal or lysosomal membranes [48].

Antibacterial activity-possessing B2M was initially observed in human amniotic fluid by Kim et al. [14]. Such activity was observed to be present in potassium buffer at pH 7.2, where B2M dissipated the transmembrane potential without disrupting the membrane. This finding indicates that an acidic pH plays a key role in the formation of B2M fibrils that cause cytotoxicity but does not function as a necessary condition for B2M to be an AMP. Nonetheless, this discovery had been seldom reported until Holch et al. who demonstrated that B2M exhibited pH-dependent antimicrobial activity [25]. Specifically, they demonstrated that B2M damages the bacterial membrane under an acidic pH. Using a radial diffusion assay for screening of the prepared B2M at acidic pH, that study found that the antimicrobial activity of B2M was present (at a fraction of its strength) in the supernatant but not in the insoluble fibrils. This finding suggests that the soluble B2M containing monomer and oligomers, and perhaps also the B2M fragment, may contribute to bacterial zone inhibition. Such a contribution may also be due to the decrease in pH that drives β-sheet structural exposure to the molecular surface and leads to the formation of B2M-related fibrils [47]. DRA is associated with an increase in B2M among patients with renal failure who persistently receive blood dialysis. B2M-related amyloidosis has one striking feature that is primarily limited to the skeletal muscular system [11]. In addition, a higher level of B2M is a systemic factor that negatively regulates cognitive function and neurogenesis in the brain [12]. This finding may also prompt us to speculate whether intact B2M or B2M-derived fibrils also play protective roles in damaged tissue. Because the epithelium represents the main interface for host defense against microbes, some defense molecules are secreted by a pH-dependent proteolytic mechanism. In a previous study, a panel of modified B2M was found in the serum of patients who had one of a variety of diseases, including small cell lung cancer [49], and in the synovial fluids of patients with rheumatoid arthritis. Although a limited proteolysis of B2M revealed that the truncated species exhibits a higher tendency to self-aggregate than the native form does [50], the mechanism underlying the shedding of B2M remains largely unclear. Whether such fragmented B2M shedding plays a role in innate defense must be investigated.

**Table 2 ijms-22-06330-t002:** Human and synthetic AMPs exhibiting pH-dependent antimicrobial activity.

Source	AMP	Antimicrobial Activity at Acidic pH	References
Human	B2M	Increase	[25]
Human	LL-37	Decrease	[51]
Human	β-defensin 3	Decrease	[51]
Human	β-microseminoprotein	Increase	[52]
Human	Hepcidin-20, -25	Increase	[53]
Human	Lysozyme	Decrease	[54]
Human	Lactoferrin	Increase	[55]
Human	Psoriasin (S100A7)	Increase	[56]
Human	Phagocytin	Increase	[57]
Human	Dermcidin-1L (DCD-1L)	Increase	[58]
Human	Hemoglobin β subunit (aa 112–147)	Increase	[22]
Human	Calprotectin	Decrease	[59]
Synthetic	A cationic, amphiphilic random copolymer	Decrease	[60]
Synthetic	Histidine-rich peptides (histatin)	Increase	[61]

## 4. Role of Aggregated B2M in Antimicrobial Activity

A study reported the binding of IgG to staphylococcus and to the gram-positive streptococcus groups A, C, and G [62]. The role of a potential AMP for B2M was also reported by Kronvall et al.; specifically, they demonstrated that only aggregated B2M with a molecular weight higher than 100,000 Da binds to the surface protein structure in streptococcus groups A, B, and G [63]. Apparently, monomeric B2M exhibits no binding effect. Notably, the effect of aggregated B2M on group A1 almost reached maximum binding after only 10 min of incubation. Considering the structural similarity between B2M and the constant (Fc) domains of human IgG and given that IgG binds to staphylococcal protein A, scholars have also investigated the binding effect for B2M on the bacteria. We previously observed that sB2M-9 instead of intact B2M had an SA-binding effect, with no crossing reaction for the IgG Fc domain [20]. In addition, we demonstrated that macrophage-activating lipopeptide-2 (MALP-2) but not lipopolysaccharide (LPS) induces sB2M-9 shedding, which is the same effect as that induced by SA instead of *Klebsiella pneumonia* (a gram-negative strain), implying an affinity for Fc-binding in gram-positive bacteria. Eventually, this effect triggers monocytes migration to the bacterial clumps. AMPs can self-aggregate to facilitate their insertion into bacterial membrane through a pore-forming mechanism [64]. Because sB2M-9 shedding is mechanistically similar to that of other AMPs, whether the sB2M-9-mediated effect on SA binding is associated with a monomer, a dimer, an oligomer or highly ordered amyloid fibrils remains unknown and requires further study. 

In essence, a variety of potential factors can contribute to the formation of B2M amyloid fibrils, including the concentration of free-form B2M, an acidic pH, the truncation and cleavage by proteolysis, sequence mutation, the presence of metals (especially those with a divalent cation), posttranslational modification (PTM), and lipid interaction [65]. These factors have distinct effects on the beta-sheet structure that mediates the formation of B2M amyloid fibrils. Under physiological conditions, the concentration of B2M is low in plasma serum but its level is increased in a wide range of diseases, including malignancies [66], chronic inflammation, and DRA [67]. Therefore, B2M is capable of achieving different conformations under pathological conditions. A model of acidic pH-induced molecular shedding that drives the formation of B2M amyloid fibrils was proposed [47]. Thus, intact B2M or B2M fragments may form their respective shape through aggregation, such as oligomerization, and through amyloid fibrillation under an acidic pH [68]. Without the presence of amyloid seeds, such as Cu^2+^ [69], amino acid substitution or truncation occurs through proteolytic cleavage, and the intact soluble B2M remains with no cytotoxicity to the cells. Notably, the PTM of B2M, particularly glycation, may yield amyloid fibrils that are deposited in patients with hemodialysis-associated amyloidosis [70]. The effect of B2M amyloid fibrils on cell membrane damage is modulated by the anionic components of lipids. Thus, this phenomenon suggests that B2M amyloid fibrils in the trafficking pathway through endocytosis at the acidic compartment may play an important role in the diseases in question. Moreover, the endocytosed B2M amyloid fibrils can disrupt endosome and lysosome membranes, eventually leading to the necrosis and apoptosis of rabbit synovial fibroblasts [48]. Thus, B2M amyloid fibrils that disrupt lysosomal membrane to inhibit protein degradation by lysosomes may contribute to amyloid fibrils-mediated pathogenesis [48,71]. The mechanism of B2M underlying the pathogenesis of ADR is indicated by pore formation, a similar feature to those of other amyloid-related diseases [68]. Although amyloid fibrils, such as amyloid beta-protein (Abeta), have long been believed to act as a key mediator of amyloidosis, they also exert antimicrobial activity against a variety of microorganisms [72,73]. In a recent study, Hoch et al. reported that the antimicrobial activity of respiratory B2M is pH-dependent due to the formation of B2M amyloid fibrils [25]. Only aggregated B2M with Mw > 100,000 Da can exert considerable antibacterial activity. This effect is based on the preparation of aggregated B2M at pH 4.5 in vitro. Such antibacterial activity is dramatically decreased when the prepared B2M has an almost neutral pH. Tipping et al. also reported that intact B2M in monomers is nontoxic at a pH 7.4 [47]. The molecular mechanism underlying the cytotoxicity of B2M amyloid fibrils is mostly associated with the membrane disruption of target cells. The aggregation of B2M into amyloid fibrils that induce membrane disruption involves an acidic pH and the modulation of endosomal lipids [70]. This phenomenon suggests that B2M amyloid fibrils cause cytotoxicity through the interaction of amyloids and lipids in the endocytic pathway under acidic condition. Relative to the induction of cytotoxicity by B2M amyloid fibrils at acidic pH, the antibacterial action of B2M at neutral pH, as reported by Kim et al., caused no observable cell membrane disruption [14]. Thus, a physiological role of antimicrobial activity of B2M in innate defense is similar but not identical to the disruption of pathological membrane by B2M-mediated amyloid fibrils at abnormally low pH.

Because a beta-sheet rich molecule such as B2M exhibits biological effects that differ depending primarily on the concentration and pH environment, a sB2M-9 concentration of 5 μg/mL is present in the culture medium of human REC, which is close to that in human nasal fluid and other biological tissues [20,74]. This concentration is also similar with those of other AMPs that exhibit channel forming activity [75]. Caution must be exercised when interpreting in vitro finding obtained using doses that are higher than their physiological counterparts and findings obtained under acidic pH, which enables B2M to disrupt microbe membranes. This finding seems to indicate the amyloid-mediated cytotoxicity of B2M fibrils rather than the effect of AMP [11,25]. Surprisingly, when commercial B2M (from human urine) was analyzed through SDS-PAGE, followed by a more sensitive silver staining, a tiny band of approximately 9 kDa was observed and later identified as sB2M-9. Therefore, the contribution of B2M to antibacterial activity might be partly due to the sB2M-9 that co-exists with B2M in the biological sample. B2M is expressed in an intact form that can be a precursor of the cleaved fragments with different biological effects. Therefore, under different physiological conditions, sB2M-9 could be further processed using an unrecognized reduction reaction to shed the high-pI, true AMP to, in turn, induce antimicrobial activity. A study also indicated that fragmented B2M fibrils cause greater membrane disruption than their native counterparts [76]. Whether B2M or B2M-derived peptides in oligomers or amyloid fibrils exhibit antimicrobial activity remains undetermined. Nevertheless, when the effects of different states of aggregation on cytotoxicity are compared, monomers and fibrils exhibit less cellular toxicity than middle-size aggregates do. This implies that oligomers trigger cytotoxicity by being more active, as indicated by Abeta peptide and alpha-syncuclein [77,78]. Furthermore, other studies have indicated that the formation of amyloid fibrils can play a role in protection against diseases. For example, the formation of an inclusion body protects Huntington’s disease and that of a Lewy body protects against Parkinson’s disease [79]. Riek et al. discovered that amyloid fibrils serve as functional storage containing peptide hormones in pituitary secretory granules and are later dispersed into extracellular space through exocytosis [80]. Because the membranes of mitochondria and bacteria are structurally and functionally similar, amyloid fibrils also cause the depolarization of both membranes [81]. This finding suggests that the beta-sheet dominant structure of amyloid fibrils that causes cytotoxicity toward eukaryotic mitochondria may exert a quite similar effect on prokaryotic membranes. The near co-existence of two similar effects caused by AMP and amyloid fibrils indicate that antimicrobial activity and cytotoxicity can serve as a friend or foe to host cells. Thus, this double-edged sword present challenge in the development of a pharmaceutical agent.

## 5. Conclusions

In addition to sB2M-9 shedding, a fragment with a high pI value hidden in B2M that governs antimicrobial activity could be released in response to a microbial challenge. Moreover, intact or fragmented B2M, which is the more numerous, self-aggregates to form oligomers and amyloid fibrils that may either protect cells against microbes or, conversely, wreak havoc in the body if the cleavage process is dysregulated (Figure 2).

## Figures and Tables

**Figure 1 ijms-22-06330-f001:**
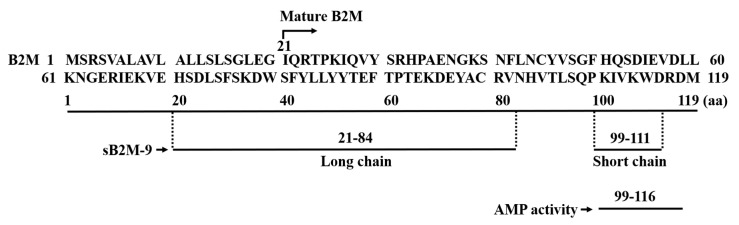
Diagram for the sequences of B2M and B2M-derived peptides with AMP activity. Mature B2M is a polypeptide of 99 amino acid residues (21-119). sB2M-9 [20] is composed of long chain (21–84) and short chain (99–111) amino acid sequences as indicated by a dashed line, cross-linking with a disulfide bond (not shown). Sequence 99–116 (theoretical pI = 10.06) with AMP activity [25] is shown in comparison with the short chain sequence (99–111, theoretical pI = 9.51) deduced from sB2M-9. One-letter code is used for the amino acid sequence of the peptide.

**Figure 2 ijms-22-06330-f002:**
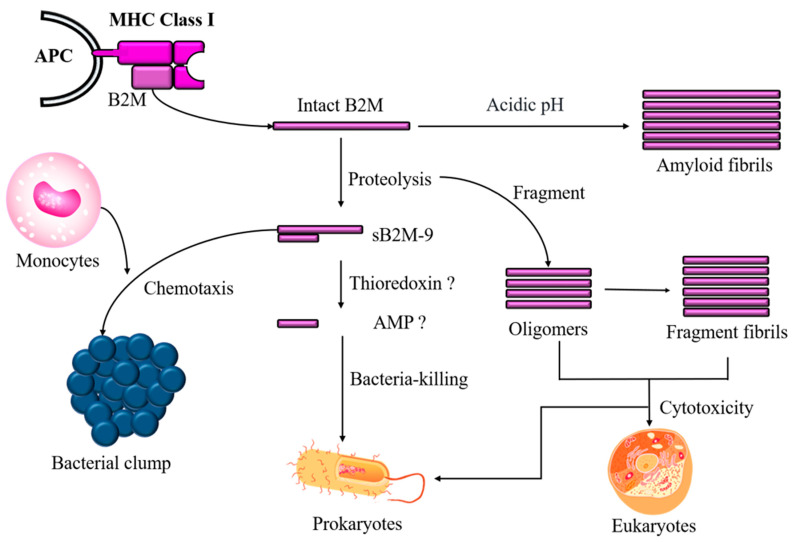
Diagram for B2M as a precursor of sB2M-9 chemokine and perhaps as a potential AMP. B2M that is at an increased level at either the intact or fragmented state under an acidic pH is prone to structural aggregation and possibly into amyloid fibrils that cause cytotoxicity and membrane disruption in eukaryotic cells; this effect is similar to that of AMP, which usually forms pores on microbe membranes.

**Table 1 ijms-22-06330-t001:** Shot-gun proteomic analysis of a panel of cationic polypeptides in a culture medium of REC. A549 cells, a type II alveolar epithelial cell line from human adenocarcinoma, were stimulated by Il-1β (1 ng/mL) following a previous method [20]. MS/MS data were processed in MassLynx 4.0 software to obtain the Mascot search results. In addition to some immune response regulators, thioredoxin is notably indicated to involve a reduction process for short chain production.

UniProt Entry	Protein Hit	Cytokine/Chemokine/AMP
VIME_human	Vimentin	Modulating cytokine [32]
ALBU_human	Serum albumin	Inducing chemokine synthesis [33]
IBP1_human	Insulin-like growth factor-binding protein 1	None
CXCL6_human	C-X-C motif chemokine 6	Chemokine/AMP [34]
THIO_human	Thioredoxin	Chemokine [30]
CXCL5_human	C-X-C motif chemokine 5	Chemokine [35]
TIMP1_human	Metalloproteinase inhibitor 1	MMP1 inhibitor: Regulating AMP shedding [36]
TPM1_human	Tropomyosin alpha-1 chain	None
TPM3_human	Tropomyosin alpha-3 chain	None
TPM4_human	Tropomyosin alpha-4 chain	None
K2C8_human	Keratin, type II cytoskeletal 8	None
IL6_human	Interleukin-6	Cytokine/Chemokine [37]
TMEM2_human	Transmembrane protein 2	None
B2MG_human	β-2-microglobulin	Chemokine/AMP [20,25]
LUZP1_human	Leucine zipper protein 1	None
IL8_human	Interleukin-8	Cytokine/chemokine/AMP [27,38]
FETA_human	α-fetoprotein	A modulator of the pro-inflammatory response [39]
IBP6_human	Insulin-like growth factor-binding protein 6	Chemokine [40]
CH10_human	10 kDa heat shock protein, mitochondrial	None
REL_human	Proto-oncogene c-Rel	Cytokine regulator [41]
DEST_human	Destrin	None
COF2_human	Cofilin-2	None

## Data Availability

Not applicable.

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
