# Peer review of "The Double-Edged Sword of Beta2-Microglobulin in Antibacterial Properties and Amyloid Fibril-Mediated Cytotoxicity"

_ijms, 2021, doi:10.3390/ijms22126330_

Round 1

Reviewer 1 Report

This is an interesting review on an emerging field of research: Antimicrobial activity of amyloid-forming peptides. Insofar, the topic is well taken and of potential interest to the readership. Concerns are primarily due to the style of the paper.

  1. The title should be changed: the current title suggests a research paper.
  2. There is a need for linguistic refinement. Especially chapter 2 must be extensively revised.
  3. The citation list must be checked again, for instance, there is a duplication of reference 12 (again cited as 52). 

Author Response

Author's Reply to the Review Report (Reviewer 1)

Open Review

English language and style

( ) Extensive editing of English language and style required
(x) Moderate English changes required
( ) English language and style are fine/minor spell check required
( ) I don't feel qualified to judge about the English language and style

Is the work a significant contribution to the field? 3/5

Is the work well organized and comprehensively described? 3/5

Is the work scientifically sound and not misleading? 4/5

Are there appropriate and adequate references to related and previous work? 4/5

Is the English used correct and readable? 2/5

Comments and Suggestions for Authors

This is an interesting review on an emerging field of research: Antimicrobial activity of amyloid-forming peptides. Insofar, the topic is well taken and of potential interest to the readership. Concerns are primarily due to the style of the paper.

Comment 1: The title should be changed: the current title suggests a research paper.

Response: Thank you for the kind suggestion. The title can be changed from the original title: “Deciphering the antibacterial properties of beta2-microglobulin” to the new one: "The double-edged sword of beta2-microglobulin in antibacterial properties and amyloid fibril-mediated cytotoxicity.” This new title with much broad view should fit well with the context of this manuscript.

Comment 2: There is a need for linguistic refinement. Especially chapter 2 must be extensively revised.

Response: The full manuscript has been edited by a native English-speaking professional. We also mention it in the Acknowledgement: We thank Wallace Academic Editing for the correction of English writing.

Comment 3: The citation list must be checked again, for instance, there is a duplication of reference 12 (again cited as 52). 

Response: Thank you for your attentive checking. We feel sorry for the mistakes from EndNote in the references processing. These mistakes have been corrected.  

Reviewer 2 Report

The manuscript explores a very interesting topic, with a clear applicative interest. However, the English form is often so confused to force the reader to reconstruct the concepts. This makes reading cumbersome and ineffective. The organization of the text also seems confused: the various points addressed cannot be clearly identified. Paragraphs are too long (line 91 to 163 cover a wide range of points, but they are hard to be intercepted. 

The conclusion is that the manuscript does not appear in focus and, therefore, in my opinion is not ready for publication.

Two further, minor comments: the number of references seems to be a little too limited for a review on the selected topic and, last but not least, the editing should be revised with more attention. For example, on line 56, the sentence is unexpectedly truncated.

Author Response

Author's Reply to the Review Report (Reviewer 2)

表單的頂端

表單的底部

表單的頂端

Review Report Form

Open Review

English language and style

(x) Extensive editing of English language and style required
( ) Moderate English changes required
( ) English language and style are fine/minor spell check required
( ) I don't feel qualified to judge about the English language and style

Is the work a significant contribution to the field? 2/5

Is the work well organized and comprehensively described?  1/5

Is the work scientifically sound and not misleading? 2/5

Are there appropriate and adequate references to related and previous work? 1/5

Is the English used correct and readable? 1/5

Comments and Suggestions for Authors

Comment 1: The manuscript explores a very interesting topic, with a clear applicative interest. However, the English form is often so confused to force the reader to reconstruct the concepts. This makes reading cumbersome and ineffective. The organization of the text also seems confused: the various points addressed cannot be clearly identified. Paragraphs are too long (line 91 to 163 cover a wide range of points, but they are hard to be intercepted. 

Response:

  1. First of all, linguistic refinement had been assisted by a native English-speaking professional. We mention it in the Acknowledgement: We thank Wallace Academic Editing for English correction. We hope this manuscript should become more readable to public.
  2. Concerning reviewer’s “confusion with points addressed can not be clearly identify”, it seems that the main challenge is to interpret our proteomic data linking with sB2M-9 and to propose a still unknown reduction reaction mediated by thioredoxin to produce a small antimicrobial peptide (i.e. B2M99-111). As Holch A et al (2020, Virulence, reference 25) demonstrated by using synthetic peptides to shed light on the antimicrobial activity for B2M99-116 that fits well with the fragment (B2M99-111) from the reduction of sB2M-9, thus we make a link (but still remains a question mark in the figure 2) between these two issues and propose if it exists possibly a natural but still unknown B2M-derived antimicrobial peptide. We think this hypothesis maybe represent one of the points to address the role for B2M with antimicrobial property. Concerning this part, we had made a bit modification to express this connection.
  3. For the part covering lines 59-167 in the second chapter, we had made it a modification with paragraphs. In these paragraphs, 1. initially we try to explain that the majority of antimicrobial peptides are released by proteolytic mechanisms, including LL-37, sB2M-9, and others. Then, we return to our finding of sB2M-9 as an antibacterial chemokine, a polypeptide with bacterial inhibition but not truly killing of microbes (refer to reference 20). sB2M-9 with chemotaxis may disclose previously unrecognized aspects of B2M in the innate defense against M. tuberculosis. Besides, we also review a previous study (refer to reference 23) that deals with a putative role for B2M against microbes. However, this molecule is not related to intact B2M but a fragmented B2M, with a molecular mass similar with sB2M-9. Here, we give it a comment by viewing their data and techniques and warn readers who should interpret antimicrobial peptides with cation. 3. We also review a recent paper by Holch et al (refer to reference 25) that deals with B2M exerts pH dependent antimicrobial activity. They use synthetic peptides to demonstrate that the antimicrobial activity of B2M localize to its c-terminus end (i.e. B2M99-116), quite similar with the small fragment (i.e. B2M99-111) of sB2M-9 if it is reduced by a reduction reaction. 4. Then we give examples of chemokine-derived antimicrobial peptide, such as CXCL880-89 (refer to reference 27) to show a peptide with antimicrobial activity largely hidden in the precursor structure, implying sB2M-9 as a precursor of B2M99-111. 5. Furthermore, when we analyze a panel of cationic peptides in human nasal fluid, we propose that thioredoxin could be a factor to involve the shedding of B2M99-111 as other reports also indicate that reduction reaction by thioredoxin plays a role in the enhancement of hBD-1 against microbes. Together, the above hints suggest a natural B2M-derived antimicrobial peptide existing is possible (see the middle in figure 2). Indeed, we also demonstrated that the B2M in intact form appears no antibacterial effect on the microbes. Thus, B2M with antimicrobial activity is still ambiguous. In this manuscript, we try to decipher antimicrobial properties of B2M from the limited literature.

Comment 2: The conclusion is that the manuscript does not appear in focus and, therefore, in my opinion is not ready for publication.

Response: As the reviewer 1 suggest that the title for this manuscript should be changed to a form for review article, now it is changed to “The double-edged sword of beta2-microglobulin in antibacterial properties and amyloid fibril-mediated cytotoxicity”. We thought that the conclusion for this manuscript is quite clear. The conclusion focuses on antibacterial properties of B2M that refers to figure 2. There are three aspects concerning this issue. Firstly, it deals with chemotactic effect for sB2M-9, derived from B2M, as an antibacterial chemokine (i.e. left part of figure 2). Secondly, it is also the viewpoint from mainstream that antimicrobial peptide hidden in the parent structure can be maturated by proteolysis. Based on our current data from proteomic analysis, we propose that a reduction reaction could be involved in the shedding of an even smaller peptide (i.e. B2M99-111) with truly antibacterial activity. This small peptide had been previously synthesized and demonstrated to display antibacterial activity by Holch A et al (2020, Virulence, reference 25). However, this small peptide still remains unknown in the nature. Thus, a diagram in figure 2 is to illustrate that B2M-derived antimicrobial peptide could exist in the nature. Therefore, question marks are indicated in the diagram (i.e. the middle part of figure 2). Thirdly, other groups also showed that B2M in aggregation and even amyloid fibrils causes cytotoxicity at acidic pH. In contrast, some other groups demonstrated that B2M in fragmented fibrils, but not full-length amyloid fibrils cause more toxic to mitochondria of mammal, and display killing effect on microbes (i.e. right part of figure 2). All these issues are discussed in chapters 2-4.

Comment 3: Two further, minor comments: the number of references seems to be a little too limited for a review on the selected topic and, last but not least, the editing should be revised with more attention. For example, on line 56, the sentence is unexpectedly truncated.

Response:

  1. Indeed, dozens of papers mentioned that B2M could display a potential activity against microbes, other than its role as just an adjuvant for MHC class I. However, it seems that only very few of them, including the one from our work, truly demonstrated the defense role for B2M against microbes. The growing evidence of B2M with antimicrobial activity, either direct or indirect, prompts us to decipher its property by reviewing a series of references in B2M as well as other antimicrobial peptides from which we learn the definition of a true defense peptide. Although, we intend to focus major in B2M, we also use other examples to dissect how B2M could be a potential peptide against microbes. Still, B2M with antimicrobial activity is still ambiguous. It needs to be verified with caution, such as whether the structural form of B2M in intact or fragmented or even amyloid fibrils status are discussed in this manuscript. Thus, this manuscript is present as a review, comment, and the most importantly to propose our point of view in this issue based on the recent comprehension about B2M as an antimicrobial peptide. Probably, these are the reasons to make confusion for reader. As the title of this manuscript has been suggested to change as “The double-edged sword of beta2-microglobulin in antibacterial properties and amyloid fibril-mediated cytotoxicity”, it could be clearer and fit well with its context and more readable.

  1. It was our mistake that the last sentence of the chapter 1 should be “………alternative defense against diseases is discussed as below.” with just a word ”below” missing. We have made it correct. Besides, we asked Wallace Academic Editing for the correction of English writing that is mentioned in the Acknowledgement.

表單的底部

Round 2

Reviewer 2 Report

The manuscript is now significantly improved and it is worth publication